# Strategies for Gaining Full Citizenship in the First Generation of Indochinese Students

Brice Fossard 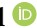

International History Center and Political Studies on Globalization, University of Lausanne, CH 1015 Lausanne, Switzerland; fossard.brice@neuf.fr

**Abstract:** The history of the acquisition of French citizenship by Indochinese university élites remains yet to be written because few researchers have looked at the role played by sport and physical education in developing the Vietnamese élite. These young students discovered such physical activities at school and many of them claimed judicial/legal equality with the French. This article will demonstrate that sports and physical education were the key stages in a strategy for certain Indochinese students to become French citizens. At the same time, this tactic generated much tension within the Vietnamese student community between the two world wars.

**Keywords:** Indochina; citizenship; sport activities; education; political tensions

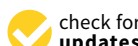



The formal creation of Indochina took place in 1887; however, it proved difficult for the French colonizers to establish themselves and transform the indigenous population into allies for France. Both urban and rural communities witnessed frequent uprisings, supported by political leaders based in other countries (Hémery 1990), such as Cuong De in Japan (Brocheux and Hémery 2007). Many Vietnamese students were drawn to Japan at the turn of the twentieth century, when the reforms of the Meiji era were creating a country as modern as France or Great Britain; some of these students thought at the time that by drawing inspiration from Japanese successes, Vietnam could recover its former power. Other countries such as China or Siam also influenced Indochinese students and intellectuals—an influence which France considered to be incompatible with the maintenance of its presence in Indochina (Cooper 2004).

The education system imposed by France, one of the justifying arguments for its civilizing mission, was developed at the time as a tool to maintain and strengthen its domination over the colonized populations (Cooper 2004). It was clearly designed to create new élites faithful to metropolitan France, to achieve better administration of the conquered lands.

Those élites in search of a model capable of transforming their country initially accepted French culture. This strategy, they hoped, would enable them to attain positions of responsibility within the administration and/or to win over parts of the market thanks to contracts signed with the French. Once associated with the running of Indochina, they demanded citizenship, "a collection of rights and obligations which give individuals a formal legal identity" thus winning equality with the French. Around three hundred Indochinese managed to obtain naturalization—but this status remained a privilege reserved to a very small minority. Besides, although Albert Sarraut had promised in 1919 to associate them more closely to the management of their country, so they would be ready to run it on their own after the departure of the French (about ninety years later), the members of those élites were still not considered as French citizens in the 1920s. Their expectations were disappointed and their demands evolved into increasingly hostile claims towards France.

The élites under consideration here were university graduates, and it seems that such education was the ideal path to achieving citizenship. Abundant historiography on the theme of colonial education shows us the milestones reached along the way by these actors yearning for a new political status. Nicola Cooper has studied the colonial manuals and instructions to identify the objectives and strategies of the French administration:

Pascale Bezançon has delivered a comparative analysis of the educational systems in the five Indochinese territories, and demonstrated the influence of western reforms on these societies (Bezançon 2002). The works of Vietnamese historians are now numerous: Trinh (1995) has analyzed the evolution in attitudes associated with western references unknown in Indochina; Nguyen Thuy Phuong (2017) has retraced the history of colonial education which became the basis for the system established in the Republic of North Vietnam; and Tuan (2016) has described the development of the School of Medicine and Pharmacy in Hanoi, the place which brought together the intellectual élites of Indochina.

The history of the acquisition of French citizenship by Indochinese university élites has yet to be written. The available data is sparse and gives only a very uncertain view of the significant sequence of events related to the motives of applicants, the number of examined files and the number of new recipients. The legislative texts establishing this acquisition of nationality are numerous and are shaped by the concern to reserve this right for an increasingly small minority; the number of such texts expanded greatly between 1880 and the 1930s, substantially modifying the conditions of access for Indochinese applicants.

On the other hand, none of these researchers has looked at the role played by sports and physical education in developing the Vietnamese élite. The young students discovered athletic activities in school and later became members of sporting societies set up for native-born élites—activities reserved to an élite within which some members claimed judicial/legal equality with the French. We will show that the creation of sports clubs, patterns of physical competition and a new sociability constituted the key stages of a strategy developed by some Indochinese students whose ambition was to become French citizens; conversely, other sportsmen were not attracted by this lure of smoke and mirrors and tried to create a Vietnamese citizenship. French citizenship was either an objective to be attained or a dead-end for one segment of these new intellectual élites.

The basis for our demonstration lies in the files of institutional archives and nine newspapers analyzed in Cambodia, Vietnam and France (National French Library, Dinan Library, History Department of the Army at Vincennes, Overseas Archives Center at Aix-en-Provence) as well as in interviews undertaken in Hanoi, Hô Chi Minh City and Vientiane.

However, it is impossible to offer a single model connecting further education and access to citizenship, since there are four different types of itinerary (which reflect the pattern of life among colonial élites): the Indochinese who sought French citizenship; those who were citizens after further education in France or Algeria; those who did not seek citizenship even though they were graduates; finally, those who were fighting for Vietnamese citizenship and undertook their studies in Moscow.

The only common factor among these activists was that they were the intellectual élite of Vietnam, but not all had the same expectations in relation to this political status; they were all sportsmen and made use of this activity as a springboard for the attainment of a new citizenship.

We base this demonstration on the files of administrative archives and on nine newspapers analyzed in Cambodia, Vietnam and France in addition to interviews conducted in Hanoi, Ho Chi Minh City and Vientiane. Besides, we have used the resources of the Overseas Archives Center in order to gain a fairly exhaustive perspective of the naturalization applications filed between 1880 and 1940: fairly complete paperwork is available for the 1890–1925 period only; as a consequence, resorting to the press proved necessary because some newspapers published the official naturalization acts for Vietnamese and Cambodians.

First of all, we will study the sociology of the young Vietnamese who became French through naturalization before analyzing their strategies for doing so; this essential step will allow us to highlight the essential and little-known role of sports in this field; then we will cross-examine the range of reasons that led them to adopt such strategy; finally, we will show that this tactic generated much tension within the Vietnamese student community between the two wars.

## 1. Who Had French Citizenship in Indochina before 1940?

Out of an estimated population of twenty million Indochinese people, some natives became French citizens before the Second World War, thus avoiding the status of natives attached to local customary law (Brocheux and Hémery 2007). An analysis of these people's profile reveals a great social and ethnic diversity: they include high-ranking civil servants, such as certain mandarins in Vietnam; interpreters such as Diêp Van Cuong, who was a teacher at the Chasseloup-Laubat high school in Saigon; lawyers such as Dinh Van Xuyen; Vietnamese or Chinese merchants, well-to-do landowners, engineers and finally students (GGI 1925). What they had in common is that they were French-speaking, were often university graduates, and agreed to work with the colonial power dominating their country (Slotfom III 1927).

The students were far from being the most numerous; however, they made up a very particular group because they embodied the future of their country. These young men and a few young women made up a very small minority in Vietnam: in fact, there were only 539 Vietnamese students (95% of the total number in 1930) at the University of Hanoi; nearly half of them studied medicine; only 21 of them read law, and twice as many attended the business school. Their education was of lesser quality than that available in metropolitan France, which frequently obliged them to go and live there; Indochinese doctors, for example, were long considered as auxiliaries to their French colleagues and not as full-fledged physicians.

The largest contingent came from Tonkin; a hundred of them were from Annam and as many from Cochinchina (Bezançon 2002). However, if we observe the number of students who became French citizens *WHEN* (our emphasis) they were still students, it is Cochinchina which dominates with 15 individuals, far ahead the other territories of Indochina (GGI 1925).

Those privileged few were France's primary allies in Indochina and were especially active in education as teachers or interpreters. Indeed, the republican school system that developed after 1880 had to train the future Indochinese élites that France needed and had to turn them into sympathizers. Intellectual and academic training was therefore an essential aspect of the association policy, like Britain with some Indian princes, because it was designed for the training of young minds to accept the colonial order and to find their place within the framework of metropolitan domination (Mangan 1992). Only the children of the élite were of interest to France according to the Governor General Albert Sarraut and their presence in the education system was exceptional (*Le Courrier* 1914).

## 2. How to Obtain French Citizenship?

The Indochinese who applied for naturalization had to be loyal to France and to master the written and spoken language. Métis children, born to a settler and an indigenous woman, became citizens in 1906 (Rolland 2014). It was therefore primarily the French-speaking native people who expressed allegiance to France who possessed the required assets to have their application accepted, namely a dual culture in addition to the two criteria mentioned above. These individuals came from the Francophone or Franco-Annamite education system, which concerned less than 8000 students in 1912. Those who could enter the university had completed the upper primary school competitive exams successfully, that is the local baccalaureate prepared in the high schools of Saigon and Hanoi or obtained the full baccalaureate in France or Algeria. In 1912 the Paul Bert secondary school in Hanoi welcomed the children of colonials and only 50 Vietnamese according to the directives of Governor General Albert Sarraut (GGI 1923). The other Vietnamese students, who were also the sons of mandarins and notables, attended the primary school of the Protectorate in which a total of 600 students were enrolled.

In 1919, the Chasseloup-Laubat High School in Saigon taught 295 European and 211 Indochinese students who would continue with higher education in Hanoi or in France (Rapport au Conseil de gouvernement 1919).

In 1913 a decree stipulated that naturalization in Indochina was individual for a native and his wife provided she agreed to change her personal status, but their children did not automatically become French. This situation changed in 1919 as the minor children of the "natives" acquired French citizenship thanks to the new Minister of the Colonies and twice former Governor General of Indochina, Albert Sarraut (*L'Écho annamite* 1921). According to the administrative archives consulted at the ANOM, three Cochinchinese became French in 1919; another Vietnamese source indicated that four people were naturalized. In any case, this new system did not immediately alter the number of people who became French citizens: those who benefited from it were still very rare. On the other hand, following the Great War, some Vietnamese soldiers were granted this change of nationality, as were their children. In short, after 1918, those who could become French were no longer only the members of the old élites or of the new ones who supported France, but also other categories of the population who gave their lives for their new homeland.

Living like the French élites in Indochina or in France was the strategy embraced by a few dozen young Indochinese, including students, to prove to the administration that they were westernized and could become French citizens.

Having attended French schools allowed them to gain modern knowledge and acquire new forms of behavior (Phuong 2017): to study the Enlightenment period and the notions of liberty, equality and fraternity; to find out everyone is an individual who can think independently without obeying their community of origin; to use one's critical abilities to compare the different political regimes; to go to the stadium and engage in physical exercise like Europeans. In a way they acted as some young Bengalis who began to play football and form clubs for regular play against British teams in India (Mason 1992).

The ideas of equality and democracy even found an outlet on the sports grounds, albeit in a completely fortuitous way before 1914. Indeed, on a sports field, sportsmen and women are equal and only differentiate themselves through the mastery of technical gestures and strategies memorized during training sessions. On the other hand, this observation, which applies to European equals, citizens in short, was not yet conceived of as part of a broader political program encompassing political, economic and cultural aspects before the Great War.

However, running a sports club was an experience in democracy for some natives. Sports created a form of equality that the law did not guarantee in Indochina (Arnaud 1991). The statutes of the associations provided that all members had equal voting tights regardless of their geographical or social origin. Of course, in Indochina this learning experience only really worked for colonials, as the Indochinese who benefited from political rights were extremely few. While we know how many people obtained French citizenship and when from administrative documents, the latter do not give any information on the date of death of new French citizens. It is therefore quite complicated to come up with a reliable figure at a given date. However, all participated in the life of the association and had the same rights and duties as the colonials. Thus, at least theoretically, a Vietnamese had as much power as a Frenchman in a sports society. Moreover, learning about Western modernity and/or democracy was possible for members of the board: Vietnamese clubs also had presidents, vice-presidents, secretaries and treasurers who learned how a modern association worked. When voting, these people had no more power than the simple members (Statuts Tanan sport 1913). The statutes of the "Tanan sport" society indicated that "the members of the board of directors are elected for one year and are eligible for re-election". The appointment of a new member could only be achieved "by a majority vote by the members of the committee". Learning about democracy (voting, respecting the majority decision) was allowed by article twenty-four: the Board of Directors "can only be overthrown by a General Assembly, after discussion, by a majority of the votes cast [...]"; article twenty-five stipulates that "the decisions of the Board are taken by the majority, with at least five of the members present". Moreover, each Indochinese person holding the position of president, vice-president, secretary or treasurer performed the same tasks and had the same responsibilities as a French person living in Indochina. Moreover, voting was

a fairly frequent act because the boards of directors were elected for only one year; anyone could stand as a candidate and vote at that time, on the sole condition that they had to send in their declaration candidacy before the general assembly. When voting, as in any metropolitan sports society, "elections were ruled by absolute majority in the first round and relative majority for the following rounds" (Statuts Tanan sport 1913). The members of the board thus discovered the complexities involved in the management of a modern society through sports.

At that time, new categories of Indochinese sportsmen (civil servants, shopkeepers, soldiers, pupils and students) wanted to participate in physical activities that embodied Western modernity and thus, by the same token, achieve the standard of living of the French colonial élite who founded the first sporting associations by 1900 in Saigon and Hanoi. As these football or tennis clubs were allowed mostly for European colonizers, some Indochinese elite members created their own clubs before 1914; very quickly, these Indochinese associations were the most dynamic by attracting students, officers, industrial workers and some peasants. These Indochinese already lived mostly as Westerners did and there prevail many signs of the transformation of their way of life: they wore suits, learned French, worked with the French élite within the colonial administrations, placed very pressing requests with the Ministry of Public Education in Indochina to send their children to the best French schools, read the Saigon or Hanoi newspapers in French in order to browse the sports articles, attended the racecourses or the soccer fields on Sundays in these two cities and clearly expressed their desire to live like the French élite thanks to modern sports.

As in Pondichery, pupils and students from Saigon and Hanoi participated in the creation of many soccer teams during and after the Great War; so the local teams were more numerous than in French West Africa by 1925 (Ruffié 2003). At that time, the Vietnamese sports pioneers came from the upper classes; being close to the French, they had all the leisure needed to discover these agonistic activities and share them with their friends and children. The latter had a model in Saigon, Nguyen Phu Khai, the son of a naturalized French Mandarin who was an arts and crafts engineer and the founder of the newspaper *La Tribune Indigène*. He was above all an excellent sportsman, as he became a second series tennis champion in 1919 (Bancel and Gayman 2002). This Cochin Chinese celebrity lived in the most influential Vietnamese circles, those who considered sports as a blessing because it helped to establish a climate of trust and peace among the different communities. (*L'Écho annamite* 1922) It was in this very relaxed climate between some Vietnamese and French people that many students played tennis on the courts of the Annamite Sports Circle of Saigon created in 1919 (*L'Écho annamite* 1920). This association recruited its members from among the Vietnamese élite and organized a tournament and a luncheon every year to bring together the players and their families. This sociability among Vietnamese was also an opportunity for the youngest associates (high school and college students) to evolve among the members of the political élite and the best local players. Those among them who stood out were favorable to the policy of Franco-Annamese association, the ideas of which were relayed by *La Tribune Indigène*, like Doctor Tran Van Don, who had obtained French nationality in 1923 and was an honorary member of the Annamite Sports Club, and the engineer Bui Quang Chieu, vice-president of the Colonial Council of Cochinchina, former president of the Alumni Association of Chasseloup-Laubat, of the Mutual Education Society and founder of a political group called the Constitutionalist Party (Brocheux 1992). Both were French-speaking, naturalized French citizens and sportsmen.

All of those people were Francophone, naturalized French citizens and athletes.

We may therefore infer, thanks to their example, that access to French citizenship had an undeniable cultural dimension: applicants were compelled to prove that they were French-speaking and lived like the colonial élites, athletic activities being proof of a cultural proximity with the latter.

When they left to study in Hanoi, the children of Saigon's élite continued to play tennis and soccer at the School of Public Works, the School of Medicine or the School of Law and Administration.

Sports were the common denominator of all the colonial organizations created for the students of Indochina. Indeed, they were supervised by a multitude of private or public associations that looked after them: the Foyer de l'étudiant annamite de Paul Monet (Paul Monet's Home for Annamite students) in Hanoi; the Association Mutuelle des Indochinois en France (Indochinese mutual student association), founded by Dr. Lê Quang Trinh and Nguyen Phu Khai; the Foyer des Étudiants indochinois catholiques (Home for Catholic Indochinese students) run by Reverend Mollat in Bourg-la Reine; finally the Maison de l'Indochine (Indochina House) in Paris, founded by the industrialist Auguste-Raphaël Fontaine. All of these study centers were also places of acculturation through sports so that students did not become bored and did not associate with radical elements hostile to France.

It seems that the condition of students was very precarious in the 1930s due to the economic crisis and to Governor General Pasquier's decision to abolish the scholarships granted to those young people, which was denounced by the Grand Council of Economic and Financial Interests of Indochina in 1933 (*La Dépêche d'Indochine* 1933). However, the young graduates who turned to the colonial civil service benefited from the creation of new positions in 1935; their assignments secured a job for them and the possibility of playing tennis or soccer on the grounds of the residences where they were posted (*La Patrie annamite* 1935).

Many of those students or very young civil servants, for example, played for Vietnam's most prestigious team, the Giadinh Star (Fossard 2021). The Hanoian public had numerous opportunities to applaud the soccer exploits of the students, who even became the champions of Tonkin in 1936 (Annam nouveau 1936). How many of these students applied for French citizenship in the 1930s? The answer to this crucial question is surely to be found in the Vietnamese archives to which we will be reporting in a few weeks. On the other hand, we know more about the Vietnamese students who made the opposite choice, those for whom sports made it possible to realize that they were proud to wear their colors, even in a country divided into three entities, and to beat colonial teams like Mohan Bagan in Calcutta in 1911 (Mason 1992). Among them, some were to make another choice for their society in 1945: the creation of a Vietnamese citizenship in a free country (Fossard 2018).

## 3. Why Apply for French Citizenship?

Around 1909, Paul Arnoux wrote that "the Cochinchinese are not interested in politics." This assertion was highly debatable at a time when the opponents of France, Prince Cuong Dê in Japan and the pirate Dê Tham in Tonkin, found support in public opinion; the fact that certain local élites asked for and obtained the right of citizenship, which is the very definition of the word "political," belies the contention. Moreover, young graduates like Bui Quang Chieu were becoming involved in politics as soon as they returned to Saigon. Applying for French nationality was an eminently political act, even more so at a time when Vietnam was breaking into several pro or anti-colonial factions embodied by Phan Chau Trinh and Phan Boi Chau (Brocheux 2011). Finally, this assertion was totally obsolete by 1925 when hundreds of young Vietnamese took to the streets in Saigon and Hanoi and tried to do away with colonial tutelage (Hémery 1990).

These young people were students at the University of Hanoi, which was an extremely rare situation for Indochinese teenagers. They and their parents belonged to the local élites who participated in the administration of Vietnam or benefited from the presence of the French. These young people belonged to the dominant group and meant to remain in it, thus they applied for naturalization. Those who were granted citizenship would be "a minute minority integrated into the colonial regime (acting as economic comprador and political yes-man)" (Brocheux and Hémery 2007).

Besides, becoming French would make it possible to break away from identification with the people defeated by this colonial power and to embody the renewal of a country

resolutely looking to the future by exploiting the possibilities of upward mobility allowed by the colonial administration. For personal or family reasons, these individuals resorted to a strategy of rapprochement with the occupying power in order to prepare their own future and/or that of their country.

French citizenship gave them political rights as well as obligations (such as paying taxes and being liable to conscription) which were the same as those experienced by the French living in Indochina.

That new status also guaranteed access to higher education for children whose fathers had been naturalized; they could travel freely to France, Algeria or study at the University of Hanoi and obtain the same diplomas as their French peers. Before the Great War, admission to the French colleges in Hanoi or Saigon depended on the goodwill of the administration; the Governor General of the time, Albert Sarraut, had decided that this was to be reserved to a restricted élite. If they were French, those children were admitted directly to French-speaking schools. Moreover, their citizenship granted them the freedom to study in metropolitan France, although the general government attempted to limit the departure of young Indochinese to France in the 1930s in order to curtail the number of "déclassés" individuals, i.e., people who returned to Vietnam with diplomas but could not find employment in the colonial civil service.

The much-coveted French citizenship also had an economic and social dimension frequently mentioned in the press.

Some of them also applied to become French citizens in order to benefit from the same rights as the colonials when applying for a position within the colonial administration: they would thus obtain an equivalent position and thus the same salary as a French colleague (*L'Echo annamite* 1921). In the 1920s, the newspaper *La Petite tribune Indigène* denounced inequalities in administrative careers where "an arbitrary racial hierarchy reigns". It then appears that naturalization was a potential solution to escape this discrimination. However, in 1939, an Indochinese newspaper noted that the situation denounced twenty years earlier had not changed: there was still no equal pay for civil servants in Indochina (*L'Écho annamite* 1939).

Finally, having French citizenship should in theory have allowed Vietnamese to vote like any other metropolitan citizen. However, Vân Thê Hôi noted that "the government grants to the most advanced Annamese a form of local citizenship, without the right to vote", that is electing the only French deputy representing Cochinchina in Paris, for example (*L'Écho annamite* 1921). This assertion must be qualified because a few thousand Vietnamese were already voters and could be elected to the Colonial Council of Cochinchina or to the indigenous consultative assemblies of Tonkin (1907), Cambodia (1913) and Annam (1920) (Brocheux and Hémery 2007). For example, from 1922 onward, the Colonial Council in Saigon was open to graduates of French and Franco-indigenous higher primary, secondary and higher education. However, these changes seemed insufficient in the eyes of some Vietnamese students who wanted to vote in the French college but not in the chambers which did not come with a decisional vote.

French citizenship was sure to open doors for certain members of the élite who hoped that it would put an end to their being demeaned by the French. Each candidate made it a personal, private endeavor which became public when the press published the official decision. At that point, the recipient entered a very restricted club, since the application of only 300 to 400 families was successful. Such change in their administrative status was seen as the crowning achievement of a process modeled upon Western standards: adopting a different sartorial style; attending secondary and higher education schools; mastering the French language; and playing the same sports as the colonial élites.

This change of nationality was much idealized because it entailed the hope of regaining one's dignity. However, French citizenship did not change the way those individuals were viewed by the colonizers: in their eyes they remained Vietnamese, they played sports among themselves and had little access to managing positions within the administration. Citizenship did allow them at least to create a legal political party, such as Bui Quang

Chieu, or to rub shoulders with certain Annamitophile colonial elites such as Governor General Pierre Pasquier or Colonel Sée, who supported the widespread practice of athletic activities in Indochina. Moreover, the Vietnamese who had become French citizens were too scarce to attract the young members of the élite who had no desire to wait for France to be more generous. Besides, access to French citizenship for all Vietnamese was never a demand of the leaders of the moderate *Jeune Annam* (Annam Youth) movement; if Nguyen Phan Long considered that naturalization should be granted to the Vietnamese élites and to them alone, he never campaigned for generalized naturalization. Bui Quang Chieu was on the same wavelength.

Faced with what looked like a delusion, more and more young Vietnamese did not even imagine whether becoming French could change their lives: in the mid-1920s, they opted for radical methods and made it known at once through street demonstrations.

## 4. A Divisive Issue for Vietnamese Students

The Indochinese youth could choose among several political directions, more or less favorable to French interests: they could support the reinstatement of Prince Cuong Dê and the departure of the French; they could follow the instructions of Moscow for the colonized peoples; they could turn to the ideas of Nguyen Ai Quoc, called Nguyen the Patriot, a countryman who spoke in their name during the Conference of Versailles; they could be inspired by the example of Chinese youth who demanded the end of Western concessions; finally, they could participate in the policy of collaboration offered by Sarraut (Phuong 2017). Each of those choices implied a different political orientation and therefore potentially different definitions of citizenship: citizenship would not exist if the monarchy was re-established; it would apply to everyone in case of a communist victory; it would remain the privilege of an elite few if France retained control of Indochina.

In the mid-1920s, not all young Indochinese had yet chosen a side, but generally their political consciousness was forged in high school, the great site of anti-colonial protest in the years 1925–1928, and at university. If radical theories were opposed to moderate ones, especially in student associations in France, this debate was not the only one that mobilized these young people. For example, at the Indochinese students' congress in Aix-en-Provence in 1927, the students attending did not formulate any project aimed at obtaining French citizenship; given the conclusions of the meeting, it is obvious that their goal was above all to improve their lives in France. For example, they wanted to create more sports associations to be as skilled as the European, American or Japanese students who liked to play basket-ball, volley-ball or rugby (*L'Écho annamite* 1927). As for some Japanese students, these Indochinese students discovered these physical activities abroad and used to practice them when they come back home. Therefore, in Indochina like in Japan, some young elite members taught that the role of sport in education was very important.

This approach aiming at obtaining another nationality was considered as a betrayal by other young Vietnamese people; those who expressed the most virulent opposition to this phenomenon were high school or university students in France. From 1925, a fraction of Vietnamese youth became radicalized and openly opposed France and its local representatives, the sovereign raised in the Paris region and the moderate pro-French bourgeoisie. In doing so, these young nationalists denounced the behavior of the Vietnamese who, according to them, changed their nationality for opportunistic reasons. These critiques were voiced out during meetings of the Indochinese student unions in metropolitan France at the end of the 1920s. The French police monitored the actions of five students "sons of rich Indochinese (...) who deeply hated France and wanted to bring about the liberation of their country" (Slotfom III 1927). In Meurthe-et-Moselle, a Vietnamese student publicly expressed his anti-colonial orientation: "We are under the yoke of the French in Indochina, but we are as civilized as they are, as intelligent and mature enough to rule ourselves." The French police then kept a watchful eye over all Indochinese students, even those who were

naturalized, demonstrating that becoming French was no guarantee of loyalty to their new homeland, according to the Ministry of the Interior in 1930.

Other Vietnamese students fought against the harmful influence that their naturalized peers would have on them. They were members of the Party for Immediate Independence, led by Nguyen The Truyen. In Toulouse this party recruited followers such as Tran Cong Vy (born 28 February 1908 in Thudaumot, a high school student):

"In order to fight against naturalization procedures as well as against the entry of Indochinese into public administrations in Indochina, a group of students led by Tran Cong Vy, director of the Home for the Association of Indochinese students, has decided to form a league against functionalism and naturalization. Its objective is also to prevent the naturalization of the Annamites and thus it derides those who betrayed their homeland by applying for French citizenship".

In 1925, the newspaper *L'Echo annamite* denounced the slowness of naturalizations, one or two per year according to them; although very few applicants were successful, "French naturalization is one of the questions that most interest the Annamite circles of Cochinchina" wrote this moderate nationalist press organ (*L'Écho annamite* 1925).

Three years later in metropolitan France, we observe that it is not the number of Vietnamese who have become French that frightened these students from Toulouse, three hundred in 65 years of French occupation according to the Annamite Echo, but rather the influence that certain personalities could have on the rest of the population: being naturalized, they pledged allegiance to the French flag and sent a clear message to their countrymen, that of loyalty to France. However, these students from Toulouse took the opposite approach by resisting the occupying power; they then targeted the models that might thwart their project of national emancipation. Moreover, some young people in Indochina used their football club and the football matches to develop nationalist propaganda from 1928 to 1939 (Fossard 2020).

This fraction of Vietnamese student youth embodied a potential threat to France, something the journalist Vân Thê Hôi had already prophesied in 1921:

"At the moment, in the midst of the Annamese people, an élite is being formed which will become numerous enough in ten years or so to lead the masses in its wake. This elite will be a powerful aid to French domination or a grave danger, depending on whether France will have had the supreme ability to incorporate it intimately or committed the irreparable blunder of making it hostile or indifferent by deliberately removing it from its orbit."

Having deliberately decided to reduce the number of new citizens to its simplest expression, France was not able to attract the sympathies of these students, who were nevertheless eager to invest in the transformation of their nation. Without a possible access to the metropolis, the young Nguyen Ai Quoc, for example, was refused entry to the Colonial School in Paris, and their plans moved to other horizons, Moscow and nationalist China for example.

## 5. Conclusions

The vast majority of the Indochinese population never had access to French citizenship and retained their status as subjects of the Republic. Those who were granted the privilege played the role of guarantors of the republican ideology of assimilation (Saada 2003); this possibility of inclusion through citizenship was so restricted that it never yielded the desired effects expected by the colonial power, namely to attract the élites and bring them closer to France. Yet, we must note that this possibility was much debated and created a rupture that could not heal in these dominated societies in search of identity and new social projects.

The first contribution of this article is to offer readers an analysis of the role of sports in the emergence of a political consciousness for some young Vietnamese. The second is to quantify the number of students who had access to French citizenship, while showing the limits of our demonstration knowing that the administration did not produce any list of

recipients between 1925 and 1945. Finally, the third contribution was to demonstrate that if being involved in sports caused very little debate in Indochina, becoming a French citizen was quite different, as it was a very divisive issue in the years 1925–1939.

The candidates to this change in nationality resorted to political, economic and social arguments to justify their request. Proving one was Francophone could also be achieved via athletic activities, the practice of which being the common point between all the personalities studied in this article.

This article is a mere progress report offering some food for thought: it needs to be completed by further investigation in the Pierrefitte-sur-Seine national archives. We hope to find there the application files of the Vietnamese with the statement of their motivations, which will allow us to validate our first conclusions and/or to bring out new avenues of investigation.

**Funding:** This research received no external funding.

**Institutional Review Board Statement:** Not applicable.

**Informed Consent Statement:** Not applicable.

**Conflicts of Interest:** The author declares no conflict of interest.

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
