# Peer review of "Strategies for Gaining Full Citizenship in the First Generation of Indochinese Students"

_socsci, doi:10.3390/socsci10040129_

Round 1

Reviewer 1 Report

 Your paper has some potential. However, I am not convinced by the main argument. By your own admission, sport emerges in the paper as only one element in the overall profile of those who were given citizenship.  Furthermore, there is insufficient contextual information about the development, characteristics and importance of sport in Indochina in the period being discussed. The points that are made about the ways in which some people were influenced, arguably more by education than by sport, to desire either naturalization or independence is interesting. However, the very existence of these differences of opinion tend to undermine the central argument in the paper as it relates to sport. If these people had shared experiences, including some engagement with sport, why did they hold very different views? How important was sport in influencing participants to formulate their political opinions or, indeed, more generally.

Author Response

I added some information about the context to explain why so many Indochinese sportig clubs were created by 1918.
Then I wrote that sport activities were sometimes a way to develop a nationalist propaganda before and after the matches, as I explained in a previous article quoted in the endnotes:
By playing football some young Indochinese people could meet some nationalist or members of the Communist party and could become either nationalist, or communist. Most of them remained neutral: I have never found any football club which was totally nationalist or communist; four clubs were closed from 1928 to 1939 because they gathered some Communists. The Colonial Police, la Sûreté Générale, spied on and harassed these political opponents, and the colonial administration checked all the clubs every year to ban the opponents.

Reviewer 2 Report

Thank you for sending me this interesting paper.
This paper sheds light on one of the little-known aspects of French-Indochina social history in the first half of the 20th century - the role of sport societies in formation of local elite, loyal to the French colonialists.

From our point of view, the work under review would benefit if the author pointed out the similarities and differences in the formation process of local elite in colonial Indochina in comparison with both, other French colonies and colonies of other developed countries, such as: Britain and Holland, as well as of developing countries, such as Japan, in the same period.

I believe that paper "Strategies for gaining full citizenship in the first generation of Indochinese students" would be very important to the readership of Social Sciences. 
I will be glad to review the revised manuscript. 

Author Response

I added some information about Pondichery and the French West Africa about sports in the 1920-1930s; I have compared the situation in Japan with what some young Indochinese students have learnt in France and did when they come back at home; then I added some information about the British colonial action towards some Indian princes (the association policy).

Round 2

Reviewer 1 Report

Thank you for responding so positively to the reviewers' comments.